# Health and Work Environment among Female and Male Swedish Elementary School Teachers—A Cross-Sectional Study

**DOI:** 10.3390/ijerph17010227

**Published:** 2019-12-28

**Authors:** Maria Boström, Christina Björklund, Gunnar Bergström, Lotta Nybergh, Liselotte Schäfer Elinder, Kjerstin Stigmar, Charlotte Wåhlin, Irene Jensen, Lydia Kwak

**Affiliations:** 1Unit of Intervention and Implementation Research for worker health, Institute for Environmental Medicine, Karolinska Institutet, Stockholm 17177, Sweden; maria.bostrom@ki.se (M.B.); christina.bjorklund@ki.se (C.B.); gunnar.bergstrom@ki.se (G.B.); lotta.nybergh@ki.se (L.N.); charlotte.wahlin@regionostergotland.se (C.W.); irene.jensen@ki.se (I.J.); 2Department of Occupational Health Sciences and Psychology, Centre for Musculoskeletal Research, University of Gävle, Gävle 801 76, Sweden; 3Department of Global Public Health, Karolinska Institutet, Stockholm 171 77, Sweden; liselotte.schafer-elinder@ki.se; 4Centre for Epidemiology and Community Medicine, Stockholm County Council, Stockholm 17129, Sweden; 5Department of Health Sciences, Lund University, Lund 221 00, Sweden; kjerstin.stigmar@med.lu.se; 6Skåne University Hospital, Lund 221 85, Sweden; 7Department of Health, Medicine and Caring sciences, Unit of Clinical Medicine, Occupational and Environmental Medicine, Linköping University, Linköping 581 83, Sweden

**Keywords:** organizational and social work environment, stress, common mental disorders, school, teachers, Copenhagen Psychosocial Questionnaire, Psychosocial Safety Climate scale

## Abstract

Background and objectives: Changes in teachers’ work situation in Sweden since the 1990s may have contributed to an increase in common mental disorders (CMDs) and burnout. However, there is a lack of research in this field. The aim was to describe how Swedish elementary school teachers experience their health, organizational and social work environment, and the psychosocial safety climate at the workplace, and especially differences and similarities between female and male teachers. Materials and methods: Data were collected with the COPSOQ, OLBI, UWES and PSC-12 from 478 elementary teachers, 81.0% of them women, from twenty schools. The response rate was 96.4%. Results: Teachers reported relatively good general health but experienced high stress, high work pace and emotional demands, low influence at work and a poor psychosocial safety climate. These factors were especially prominent among female teachers. Both women and men experienced good development possibilities and high work engagement. Conclusions: The results of this study can help us to develop a more sustainable work environment for female and male teachers. A more sustainable work environment might attract more people to the profession and incentivize existing teachers to remain in the profession.

## 1. Introduction

An important prerequisite for a sustainable working life is the management of organizational and social risk factors in the workplace. Inadequate management of these risks can result in ill health, such as common mental disorders (CMDs) and burnout, and consequently sick leave. The prevalence of CMDs, which include depression and anxiety, is alarmingly high across many professional groups and countries and are also associated with extremely high costs for society [1]. Several reviews and independent studies from both Europe and Australia have observed that teachers are an especially vulnerable professional group [2,3,4,5,6,7,8,9,10]. In Sweden, teaching has been identified as one of the professions with the highest prevalence of CMDs and burnout [11]. Teachers also have the second highest sickness absence rate of any professional group in Sweden, with rates higher for women than for men [12]. An urgent need therefore exists for appropriate prevention and intervention strategies in the workplace for this professional group. These strategies need to take the differences between female and male teachers into account. The teaching profession in Sweden is female dominated with an overall distribution of female teachers in elementary schools of 75.7% [13]. A first step in identifying and developing appropriate strategies is to gain up-to-date knowledge about how teachers perceive their organizational and social work environment. 

We need a better understanding of the prevalence of risk and protective factors for CMDs and burnout in teachers’ organizational and social work environment. According to the job demands–resources model, risk and protective factors can be related to psychological job demands (workload, work pressure) and job resources (control over work tasks, social support from colleagues and supervisors) [14]. Several studies looking specifically at teachers have identified associations between these work-related organizational and social factors and CMDs and burnout. Identified factors include high job demands caused by the high emotional demands of interaction with pupils and parents [15,16,17,18], as well as high job demands caused by long periods of concentration and divided attention [2]. High workload resulting from a heavy administrative burden which leads to a lack of time for teaching has also been identified as a risk factor among teachers [19]. Studies have also shown that teachers lack social support from supervisors [20] and from colleagues [21]. This, when combined with poor leadership [10], can also result in CMDs and burnout, as can lack of recovery [22]. Recovery refers to those activities that give individuals energy and capacity to deal with their job [23]. Lack of recovery has been shown to partially mediate the relationship between work stress and burnout in teachers [7,24]. According to a cross-sectional study conducted in Sweden, teachers’ non-curriculum time (e.g., time for preparation and administration, which is often conducted at home) has been identified as having a major impact on their work–life balance [25]. Studies of work-life balance have shown that an imbalance between work and multiple life roles is a risk factor for CMDs [26]. Further, gender egalitarianism has been shown to moderate this relationship. In other words, in cultures with less adherence to traditional gender patterns, a work–life balance is likely to have a positive effect on mental health [27]. Despite all we know about the relationship between work-related psychosocial factors and CMDs among teachers, there is surprisingly little quantitative data about how teachers experience their job demands, job resources, recovery and work–life balance. 

Because there are observable differences in the prevalence of CMDs and burnout, and related sickness absence between female and male teachers, prevalence studies should include comparisons of how female and male teachers perceive their organizational and social work environment. Such comparisons will not only help us to understand the difference in the prevalence of CMDs and burnout, and related sick leave but will also provide information about whether we may need different preventative strategies for women and for men. Including a gender aspect in research questions has been raised in research into occupational health [28] and burnout [29]. However, few previous studies of teachers have included this type of gender focus [8,9,10,15,21,30,31]. 

An overview of teachers’ perceived organizational and social work environment should also include information about how teachers perceive the psychosocial safety climate at their workplace. The psychosocial safety climate is a relatively new concept defined by its developers as “policies, practices, and procedures for the protection of worker psychological health and safety” [32]. Many studies have shown that the psychosocial safety climate is related to work demands and work resources [32,33]. Studies have also shown that the psychosocial safety climate is negatively associated with burnout and mental ill health [34,35]. Finally, longitudinal studies have shown that the psychosocial safety climate can have a protective effect: when workplaces have a high psychosocial safety climate, work demands and mental ill health are low [32,36,37]. Studies of the psychosocial safety climate in schools have shown that a higher perceived psychosocial safety climate is associated with lower job demands. Moreover, the psychosocial safety climate has also been shown to have a significant effect on fatigue and engagement, when job demands have been controlled for [32,37,38,39]. Most studies examining the psychosocial safety climate have been conducted in Australia. To our knowledge, no previous studies have examined the psychosocial safety climate among teachers in Europe. 

The aim of this study was to describe how Swedish elementary school teachers experience their health, organizational, and social work environment, and the psychosocial safety climate in the workplace. A further aim was to explore the differences and similarities between how female and male teachers experience these factors. 

## 2. Materials and Methods 

### 2.1. Study Design

This is a cross-sectional study that is part of a larger randomized controlled implementation study. The overall aim of the implementation study is to develop and evaluate implementation strategies which will help school principals to implement guidelines for the prevention of CMDs in the workplace [40]. Detailed information about the project, a 24-month, cluster-randomized, waiting-list-controlled trial in a school setting, can be found in Kwak et al [41]. The baseline data of the project were used for the present study. 

### 2.2. School Context

The educational system in Sweden is based on a nine-year elementary school for children aged 6–15. Elementary schooling is divided into four stages: preschool stage (year 0), lower stage (years 1–3), middle stage (years 4–6), and upper stage (years 7–9). There are three different elementary school models in Sweden. F–3 schools cover the preschool and the lower stage, with pupils up to 9 years of age. F–6 schools cover the preschool, lower, and middle stages, with pupils up to 12 years of age. F–9 schools cover the preschool, lower, middle, and upper stages, with pupils up to 15 years of age. Most elementary schools in Sweden are municipally run and they employ teachers, recreational pedagogues, remedial teachers, teacher assistants, administrative employees, and school management. Recreational pedagogues are responsible for activities before and after compulsory school hours and during breaks. Teacher assistants support teachers during the teaching day. 

The present study involved twenty municipal elementary schools in two Swedish municipalities. Eight of the schools are located in a large city and have 200–800 pupils. The remaining twelve are located in a rural district and have 15–500 pupils. The number of staff varies between 7 and 133. The larger the school is, the higher the proportion of male employees. The schools reflect a diverse spread of socio-economic status areas in the two municipalities. 

### 2.3. Study Sample

In the period from September to October 2017, all employees of the participating schools (840 individuals) were invited to complete the baseline questionnaire. Of these, 496 were teachers.

A total of 705 people answered the questionnaire (76.0% women; *n* = 536), of whom 478 were teachers. Reasons for not participating in the study were not collected. The total mean response rate was 83.9%. Only teachers were included in the present study because we aimed for homogeneity of work tasks. All other employees (e.g., school principals, recreational pedagogues and administrative staff) were thus excluded from the sample in this study. The study sample of 478 teachers, reflecting a response rate of 96.4% of the total invited teachers, contained 387 women (81.0%) and 90 men (18.8%). One individual classified themselves as “other” and was not included in the stratified analyses. The sample included teachers, first teachers, preschool teachers, and remedial teachers in the lower stage (years 1–3), the middle stage (years 4–6), and the upper stage (years 7–9). 

### 2.4. Data Collection

Data was collected in two phases. In the first, the research team visited each school with printed questionnaires and school employees completed the questionnaire during the research team’s visit. In the second phase, employees unable to be present during the visits received an electronic version of the same questionnaire by email. Three reminders were sent by e-mail. The questionnaire included 29 questions with a total of 92 items. The questions and items all originate from existing instruments that are validated and reliability tested. A detailed description of the questions and items included in the questionnaire is given below. 

### 2.5. Questionnaire

#### 2.5.1. Individual Factors

The questionnaire asked questions which assessed sex/gender, age, and social situation (e.g., living with children). In the Swedish language no distinction is made between sex and gender, i.e., the same word is used for both gender and sex. The questionnaire asked whether participants were woman, man, or other. The third option “other” was included to allow participants to provide a response even if they do not identify as either woman or man, or if they prefer not to respond. The questionnaire also contained questions about type of profession, number of years in the current profession and at the current workplace, and frequency of working overtime (>9 hours/day, based on full-time work of 45 h/week for teachers in Sweden). 

#### 2.5.2. Health

*Self-rated general health* was assessed by the first question of the validated 12-Item Short Form Health Survey [42]: “In general, would you say your health is…?” with response on a five-point Likert scale ranging from (1) excellent to (5) poor.

*Perceived stress* was assessed by a single question [43]: “Stress means a situation in which a person feels tense, restless, nervous, or anxious, or is unable to sleep at night because his/her mind is troubled all the time. Do you feel this kind of stress these days?” Answers were given on a five-point Likert scale ranging from (1) not at all to (5) very much. 

*Recovery after some days off work* was formulated: “Do you feel you have recovered and are thoroughly rested when you start work after a few days off?” with responses on a five-point Likert scale ranging from (1) never to (5) very often [44]. 

*Negative exhaustion* was assessed with the four negatively framed items of the Oldenburg Burnout Inventory Scale, OLBI [45]. Satisfactory reliability has been found for these four negatively framed items [46]. The items concern being tired before work, longer times for rest, being emotionally drained and being worn out. The response alternatives range from (1) strongly disagree to (4) strongly agree. A higher score indicates a higher risk of negative exhaustion.

#### 2.5.3. Organizational and Social Work Environment

The validated instrument Copenhagen Psychosocial Questionnaire (COPSOQ II) [47,48] was used to assess teachers’ organizational and social work environment: demands at work, work organization and job contents, interpersonal relations and leadership, and work–life conflict. All these COPSOQ items have five response options on Likert-type scales, from (1) never to (5) always, or from a (1) very low, to a (5) very high extent.

Demands at work were assessed by three subscales: quantitative demands (four items, for example: “Do you get behind with your work?”), work pace (three items, for example: “Do you have to work very fast?”), and emotional demands (four items, for example: “Is your work emotionally demanding?”). 

Work organization and job contents were assessed by three subscales: influence at work, possibilities for development, and commitment to the workplace. The first two subscales consist of four items and the last of three, for example: “Do you have any influence on what you do at work?” (Influence at work), “Do you have the possibility of learning new things through your work?” (Possibilities for development), and “Do you feel that your place of work is of great importance to you?” (Commitment to the workplace). 

Interpersonal relations and leadership were assessed by three subscales: social support from the individual’s superior, social support from colleagues, and recognition from management. Every subscale contains three items, for example: “How often does your immediate superior talk with you about how well you carry out your work?” (Social support from superior), “How often do your colleagues talk with you about how well you carry out your work?” (Social support from colleagues), and “Does the management at your workplace respect you?’” (Recognition from management). 

Work–life conflict was assessed by four items, for example: “Do you feel that your work drains so much of your energy that it has a negative effect on your private life?” with responses ranging from (1) no, not at all to (4) yes, for sure.

The short version of the Utrecht Work Engagement scale (UWES) with three items was used to assess work engagement [49,50]. Example: “At my work, I feel bursting with energy”. The response alternatives range from (0) never to (6) always. An index was constructed of these items according to Schaufeli et al. [50], with a higher score indicating a higher degree of engagement. 

#### 2.5.4. Psychosocial Safety Climate at the Workplace

The psychosocial safety climate was assessed by the Psychosocial Safety Climate scale (PSC-12) [51], which contains four subscales. Each subscale consists of three questions. For example: “Senior management acts decisively when a concern of an employee’s psychological status is raised” (Subscale 1. Management commitment), “Senior management considers employee psychological health to be as important as productivity” (Subscale 2. Management priority), “There is good communication here about psychological safety issues which affect me” (Subscale 3. Organizational communication), and “Employees are encouraged to become involved in psychological safety and health matters” (Subscale 4. Organizational participation). The answer alternatives range from (1) strongly disagree to (4) strongly agree. All questions form an index where a higher number indicates a better psychosocial safety workplace climate. 

### 2.6. Statistical Analysis 

All analyses were performed with SPSS version 25 (IBM^®^
*SPSS*^®^
*Statistics)*. Frequency analyses were used for the description of the study sample and to obtain means and standard deviations for the variables and the indexes. 

A variety of tests were performed for the gender comparisons. The T-test was used for the numeric variables age, sickness absence, self-rated general health, perceived stress, exhaustion, and all single questions and indexes in Table 3. The Fisher exact test was used for the only categorical variable without a ranking, namely household. Mann-Whitney U tests were used for all categorical variables with a ranking. A significance limit for *p*-values of 0.05 was chosen. Because the study is descriptive, the different response categories were kept in Table 1 and Table 2 and not dichotomized. The small numbers of participants in some cells did not hamper a Mann-Whitney U test as it measures a comparison of the distribution and the sample sizes of both groups are enough.

Indexes of the COPSOQ II items were constructed with scales of 5–1 or 4–1 depending on the items. This was chosen instead of recoding the scales into 100–0 as suggested by COPSOQ. This decision was made in order to make it possible to show comprehensible means in relation to the other instruments and has been used in this way previously (Arvidsson 2016). All items were reversed, except for the two which are formulated in the opposite direction. As recommended for the COPSOQ II instrument, individuals who had answered at least half of the questions in each index were included in the analysis (COPSOQ homepage: https://copsoq.se). The same principal was used for the scale in OLBI, UWES and PSC-12.

### 2.7. Ethics

The project has been approved by the Ethical Committee of Stockholm (2017/984–31/5). The participants received a letter with detailed information about the project before completing the questionnaire. It was clearly explained that participation was voluntary and that they were free to withdraw without any explanation. They were also told that their employer would not receive any information about their participation. 

Informed consent was obtained from all participants in a separate document when they answered the questionnaire. Those who answered by email were told that they gave their informed consent by sending in the questionnaire. The collected data has been de-identified, and a Code Key is saved separately from the data. All analyses have been performed at group level so no individuals can be recognized. The results will also be presented at group level. 

## 3. Results

### 3.1. Individual Characteristics

The mean age of the study sample was 46.8 years of age (Table 1). The fact that the age range extends to 78 was due to retired teachers working as substitute teachers. In the study sample, 77.2% reported living with another adult or with another adult and children; 47.1% reported having children under 16 years of age living at home full- or part-time. In total, 24.9% of the teachers had been working less than five years in their current profession and 62.2% less than five years at their current workplace. 

No statistically significant differences were found between female and male teachers for these characteristics. In total, 84.1% of the teachers reported working full-time (45 h/week), with no differences between women and men. 6.5% of teachers responded that they never worked overtime (>9 h/day). A difference between the sexes was observed here, with 5.5% of female teachers reporting that they never work overtime compared to 11.4% of the males. 

### 3.2. Health 

In total, 81.6% of the teachers responded that they enjoyed good, very good, or excellent general health, with no statistically significant differences between the sexes (Table 2). Where perceived stress was concerned, 47.9% reported experiencing “quite a lot” or “very much stress”. 

Female teachers experienced significantly more stress than male teachers, with 20.2% of them reporting “very much stress” compared to 4.5% of male teachers. Overall, 16.8% of the teachers reported never or seldom recovering from work after a few days off; 18.3% of female teachers reported this compared to 9.0% of male teachers. Lastly, teachers reported high levels of negative exhaustion, with female teachers reporting statistically more exhaustion than their male colleagues.

### 3.3. Organizational and Social Work Environment 

The means and standard deviations for the organizational and social work environment factors are presented in Table 3. The mean and standard deviation for the total construct of demands at work was 3.6 ± 0.5. Significant differences between women and men were observed for this total construct and all its three subscales. The work pace subscale had the highest mean and was higher for female than male teachers. The emotional demands subscale demonstrated the second highest means, and again, women reported higher emotional demands than men. 

The mean and standard deviation of the total construct of work organization and job contents was 3.4 ± 0.5. No difference between woman and men was observed for the total construct. Its subscale possibilities for development had the highest mean and were similar for female and male teachers. Only the subscale influence at work demonstrated differences, with female teachers reporting having less influence at work than male teachers. 

The mean and standard deviation of the total construct of interpersonal relations and leadership was 3.4 ± 0.6. No differences between women and men were observed for the total construct or any of its subscales. 

Female teachers reported lower levels of recovery after some days off work than male teachers. For work–life conflict, the observed mean was 2.5 ± 0.9: female teachers reported more work–life conflicts than male teachers. The mean for work engagement was 5.1 ± 0.8, with no differences observed between women and men.

### 3.4. Psychosocial Safety Climate at the Workplace

The mean of the total construct of the psychosocial safety climate was 27.8 ± 9.0 (Table 3). Differences between women and men were observed for the total construct and all subscales. Overall, female teachers reported poorer psychosocial safety climate than their male colleagues, both for the total construct and for all subscales. The lowest mean was observed for the organizational communication subscale (6.1 ± 2.5; female teachers 5.8 ± 2.4 and male teachers 7.1 ± 2.3, *p* < 0.001). 

## 4. Discussion

The current study presents findings about teachers’ health, organizational and social work environment and psychosocial safety climate. Similarities and differences between female and male teachers were also investigated. 

### 4.1. Teachers’ Health

The results demonstrate that over 80% of teachers reported good to excellent general health. This is in line with findings for the general population in Sweden, which show that 84% of women and 88% of men with at least college or university education experienced good health [52,53]. Despite these ratings, however, most teachers in the present study reported feeling stressed. Comparable levels have been observed in other studies of teacher stress [54] and in studies of stress in other human service professions, such as nursing [55] and social work [56]. We also observed high levels of negative exhaustion (mean ≥ 2.25) [57], which is in line with findings of other studies assessing exhaustion among teachers [9] and could be a reaction to prolonged exposure to stress [58].

### 4.2. Teachers’ Organizational and Social Work Environment 

Stress, and by extension exhaustion, can be a result of an imbalance between psychological job demands (workload, work pressure) and job resources (e.g., control over work tasks, social support from colleagues and supervisors) [14]. The present study assessed three work demands (quantitative demands, work pace, and emotional demands) and six potential job resources (influence at work, possibilities for development, commitment to the workplace, social support from superior, social support from colleagues, and recognition from management). Overall, teachers reported high demands at work, especially high work pace and emotional demands. These findings are in line with research findings over recent decades of increased workload and pace of work among teachers [59,60,61]. The results are also in line with Swedish data showing that 84% of elementary school teachers reported a high workload [11]. High work pace and too much work have also been reported as main causes of work-related disorders among elementary school teachers in Sweden [62]. Our findings of high workload could be explained by one of the many political reforms of the Swedish educational system since the 1990s [63]. These reforms have included more detailed rules/curricula, national tests, and tougher inspections. These political reforms align with policies focusing on measurability and accountability implemented in other countries [64,65]. A recent study identified workload and time pressure as the strongest predictors of teachers’ wellbeing, underscoring the importance of intervening on these work demands in order to increase teacher’s wellbeing [66]. The political reforms could also help to explain the low influence at work observed among teachers in the present study. The reforms have led to the teaching profession becoming more regulated at national level and teachers having less autonomy. The high possibilities for development observed in our study could be the result of a national reform that was implemented in 2012 in order to create career paths for teachers. This reform introduced the professional status of “first-teachers”. Becoming a “first-teacher” is seen as an important step in a teacher’s development and status. The reform was implemented as a way of increasing the attractiveness of the profession and retaining teachers within the profession. Sweden, like many other countries, is experiencing a great shortage of teachers [67,68]. It has been estimated that in 2022 there will be a shortage of 77,000 full-time teachers, which is equal to the total number of teachers employed in municipal elementary schools in 2017.

### 4.3. Differences between Female and Male Teachers’ Organizational, Social Work Environment and Health

We observed small differences in job demands (quantitative demands, work pace, and emotional demands) and influence at work between female and male teachers. Despite the differences being small, we found that women reported higher quantitative demands, work pace and emotional demands, and lower influence at work. The differences in work pace and emotional demands between female and male teachers are in accordance with the findings of previous studies of Norwegian teachers [16,66]. We also observed more work–life conflicts in female than in male teachers. This could be a result of the above-described higher quantitative demands and work pace observed among our female teachers and their higher frequency of working overtime. Another explanation can be that women have greater responsibility for household chores than men [69]. Our findings are in line with other studies showing that women are at particular risk of work–life imbalance [70,71]. We also found that female teachers report poorer recovery from work after a few days off than their male colleagues, which could be related to the observed work–life conflict and working over-time. Both work–life imbalance and lack of recovery have been found to be associated with the development of exhaustion [7,70,71].

The above-described differences in how female and male teachers perceive their organizational and social work environment could be one explanation for the differences in self-reported stress and exhaustion between female and male teachers found in this study. Female teachers reported significantly higher stress levels and more exhaustion than their male colleagues, which is in accordance with the findings of several other studies in Europe and Canada [2,9,10,17,31,72]. Previous studies have discussed whether the double workload of women could be another explanation for the observed differences [9,17,73]. Even though Sweden is considered a gender-equality role model, gender differences in the amount of time spent on household work and childcare do nevertheless exist, with women spending more time on these than men [52], like many countries in Europe [74]. Further, the higher quantitative demands can plausibly be explained by so-called gendered work tasks at school—in other words, women and men having different work tasks because of gender-based expectations [75,76]. According to this possible explanation, female teachers are more likely than men to perform “invisible” work tasks (e.g., practical and social tasks for teachers, pupils, and parents), which accordingly result in an increase in female teachers’ workload and stress-levels.

### 4.4. A Systematic Management of Organizational and Social Risks in the Work Environment

One well-established evidence-based strategy to prevent CMDs in the workplace is managing organizational and social risks in the work environment. This includes having policies for risk management, assessing work-environment risk factors, intervening on identified risk factors in a participatory way and conducting regular follow-ups. This is recommended by several systematic reviews, regulatory frameworks (e.g., PRIMA-EF) and guidelines on the prevention of CMDs at the workplace [40,77,78,79]. In the current study, teachers rated the psychosocial safety climate of their school as low [80]. This implied that the participating schools lacked policy and procedures to actively manage organizational and social risk factors; that there was no clear communication about stress prevention; and that organizational and social risk were not regularly discussed during staff-meetings. However, it can be difficult to make direct comparisons of self-reported scales [80] between different countries and cultures, so the results should be interpreted with caution. Nevertheless, our findings concur with those of the Swedish Work Environment Authority that examined 30 percent of Swedish schools in 2013–2016 and found that nine out of ten had severe shortcomings in their work-environment risk management [81]. The observed level of psychosocial safety climate indicated by the teachers in the present study is lower than that observed for other professions in Australia [51]. Thus, there is a clear need to support schools with their organizational and social risk management in order to prevent CMDs among teachers and gain an understanding of how best to provide support. Current knowledge about this is scarce, as demonstrated by a Cochrane Review of organizational interventions in schools, which found only four studies that evaluated interventions targeting the causes of work-related stress among teachers [82].

### 4.5. Methodological Considerations

The study sample consists of 81.0% female teachers, which is in agreement with the overall distribution of female teachers in elementary schools in Sweden [13]. Our study sample reflects a representative selection of teachers with regard to the distribution of female and male teachers. The unequal distribution does however not influence the results of the present as the group of male teachers was sufficiently large for the chosen analyses. Our sample also has an equal representation of early career teachers (<5 years) and teachers who have been in the profession longer. However, the sample is skewed regarding time worked at the current workplace. Most teachers in the present study answered that they had been working at their current workplace for less than 5 years. This clearly reflects the high turnover rate of teachers that Sweden [83] and other countries are experiencing. Our sample represents two types of geographical area in Sweden and includes urban and rural, small and large schools and ones belonging to a diversity of social economic areas. However, only municipal elementary schools were included in the sample, which means that the results do not represent the experiences of teachers in private schools.

The aim of the present study was to describe the current state of teachers’ health and work environment, about which there is a lack of information. The data derives from the baseline of a guideline implementation study [41]. The results of the study will be published in forthcoming studies.

Most of the questions and items included in the questionnaire originate from well-established validated measurement instruments (e.g., COPSOQ II, PSC-12 and OLBI). Only negative exhaustion could be assessed in the present study. This limited our ability to make comparisons with other studies that have measured exhaustion with both positive and negative items. Previous studies have, however, demonstrated acceptable reliability of the items measuring negative exhaustion [46].

## 5. Conclusions

Teachers reported relatively good general health but experienced high stress, high work pace and emotional demands, low influence at work, and poor psychosocial safety climate; these factors were especially prominent among female teachers. In contrast, both female and male teachers reported high development possibilities and high work engagement. The results of this study will be a first step towards helping to develop a more sustainable work environment for female and male teachers. A more sustainable work environment will likely attract more individuals to the profession and encourage those who are already teaching to remain.

## Figures and Tables

**Table 1 ijerph-17-00227-t001:** Individual characteristics for teachers at baseline.

	Missing	All Teachers *n* = 478 ^a^		Females *n* = 387 (81.0%)		Males *n* = 90 (18.8%)		*p*-Value *
Age, mean ± sd, range, in years	9	46.8 ± 11.4 22–78		46.6 ± 11.4 22–71		47.6 ± 11.6 27–78		0.50
		*n*	%	*n*	%	*n*	%	
Age	9							0.74
18–29		38	8.1	28	7.4	10	11.4	
30–39		104	22.2	89	23.4	15	17.0	
40–49		130	27.7	106	27.9	23	26.1	
50–59		108	23.0	85	22.4	23	26.1	
60–78		89	19.0	72	18.9	17	19.3	
Household	1							0.35
Living alone		72	15.1	61	15.8	11	12.2	
Living alone with child/children		37	7.8	31	8.0	6	6.7	
Living with other adult		151	31.7	114	29.5	36	40.0	
Living with other adult and child/children		217	45.5	180	46.6	37	41.1	
Time in profession	1							0.37
<5 years		119	24.9	95	24.6	24	26.7	
5–14 years		140	29.4	113	29.3	27	30.0	
15–24 years		111	23.3	86	22.3	24	26.7	
25–34 years		64	13.4	55	14.2	9	10.0	
≥35 years		43	9.0	37	9.6	6	6.7	
Time at present workplace	18							0.30
<5 years		286	62.2	238	63.8	47	54.7	
5–14 years		101	22.0	73	19.6	28	32.6	
15–24 years		50	10.9	42	11.3	8	9.3	
25–34 years		17	3.7	16	4.3	1	1.2	
≥35 years		6	1.3	4	1.1	2	2.3	
Overtime work >9 h/day	4							**0.001**
Every day		60	12.7	55	14.3	5	5.7	
A couple of days per week		164	34.6	140	36.4	24	27.3	
One day per week		87	18.4	67	17.4	19	21.6	
A couple of days per month		69	14.6	56	14.5	13	14.8	
More seldom than a couple of days per month		63	13.3	46	11.9	17	19.3	
Never		31	6.5	21	5.5	10	11.4	

Note. ^a^ One individual answered “Other” in sex and was not included in the stratified analyses. Mean = the mean value of the total score for that instrument; sd = the standard deviation of the mean value; *n* = the number of individuals. * The *p*-values show the difference between female and male teachers, calculated by the T-test, Mann-Whitney U test, and Fisher exact test, indicating how consistent the results are with the 0-hypothesis with a significance limit of 0.05; bold figures represent *p*-values < 0.05.

**Table 2 ijerph-17-00227-t002:** Health among teachers at baseline.

	Missing	All Teachers *n* = 478 ^a^		Females *n* = 387 (81.0%)		Males *n* = 90 (18.8%)		*p*-Value *
Health								
Self-rated general health ¹, scale 1–5, mean ± sd		2.6 ± 1.0		2.6 ± 1.0		2.5 ± 0.9		0.11
Perceived stress ¹, scale 1–5, mean ± sd		3.4 ± 1.1		3.5 ± 1.1		2.8 ± 1.1		**<0.001**
		*n*	%	*n*	%	*n*	%	
Self-rated general health	1							0.11
Excellent		58	12.2	47	12.2	11	12.2	
Very good		162	34.0	124	32.1	38	42.2	
Good		169	35.4	140	36.3	29	32.2	
Fairly		79	16.6	67	17.4	11	12.2	
Bad		9	1.9	8	2.1	1	1.1	
Perceived stress	2							**<0.001**
Not at all		35	7.4	20	5.2	15	16.9	
Only a little/Slightly		68	14.3	51	13.2	17	19.1	
To some degree		145	30.5	112	29.0	33	37.1	
Quite a lot		145	30.5	125	32.4	20	22.5	
Very much		83	17.4	78	20.2	4	4.5	
Recovery after some days off work	2							**0.001**
Never		13	2.7	11	2.8	1	1.1	
Seldom		67	14.1	60	15.5	7	7.9	
Sometimes		166	34.9	139	36.0	27	30.3	
Rather often		162	34.0	129	33.4	33	37.1	
Very often		68	14.3	47	12.2	21	23.6	
Negative exhaustion, Scale 1–4 ¹, mean ± sd	10	2.8 ± 0.7		2.8 ± 0.7		2.5 ± 0.8		**0.002**

Note. ^a^ One individual answered “Other” in sex and was not included in the stratified analyses. ¹ Higher scores indicate a more **unfavorable** situation. Mean = the mean value of the total score for respective instrument; sd = the standard deviation of the mean value; *n* = the number of individuals. * The *p*-values show the difference between female and male teachers, calculated by the T-test and Mann-Whitney U test, indicating how consistent the results are with the 0-hypothesis with a significance limit of 0.05; bold figures represent *p*-values < 0.05.

**Table 3 ijerph-17-00227-t003:** Organizational and social factors for teachers at baseline.

	Missing	Scale	Interpretation of the Mean Values	All Teachers *n* = 478 ^a^	Females *n* = 387 (81.0%)	Males *n* = 90 (18.8%)	*p*-Value *
				Mean ± sd	Mean ±sd	Mean ± sd	
Total construct of Demands at work	0	1–5 ¹	High = negative	3.6 ± 0.5	3.6 ± 0.5	3.3 ± 0.5	**<0.001**
Subscales							
Quantitative demands	4	1–5 ¹	High = negative	3.3 ± 0.8	3.4 ± 0.8	3.0 ± 0.8	**<0.001**
Work pace	0	1–5 ¹	High = negative	3.9 ± 0.7	3.9 ± 0.7	3.7 ± 0.6	**0.001**
Emotional demands	2	1–5 ¹	High = negative	3.8 ± 0.7	3.8 ± 0.7	3.4 ± 0.7	**<0.001**
Total construct of Work organization and job contents	0	1–5 ²	High = positive	3.4 ± 0.5	3.4 ± 0.5	3.4 ± 0.5	0.27
Subscales							
Influence at work	1	1–5 ²	High = positive	2.8 ± 0.6	2.7 ± 0.6	3.0 ± 0.6	**<0.001**
Possibilities for development	0	1–5 ²	High = positive	4.0 ± 0.5	4.0 ± 0.5	3.9 ± 0.5	0.28
Commitment to the workplace	3	1–5 ²	High = positive	3.4 ± 0.7	3.4 ± 0.7	3.4 ± 0.7	0.82
Total construct of Interpersonal relations and leadership	3	1–5 ²	High = positive	3.4 ± 0.6	3.4 ± 0.6	3.4 ± 0.6	0.56
Subscales							
Social support from superior	7	1–5 ²	High = positive	3.2 ± 0.8	3.2 ± 0.8	3.3 ± 0.8	0.16
Social support from colleagues	3	1–5 ²	High = positive	3.5 ± 0.7	3.5 ± 0.7	3.4 ± 0.6	0.14
Recognition from management	8	1–5 ²	High = positive	3.4 ± 0.8	3.4 ± 0.8	3.5 ± 0.8	0.22
Recovery after some days off work	1	1–5 ²	High = positive	3.4 ± 1.0	3.4 ± 1.0	3.7 ± 1.0	**0.001**
Work life conflict	2	1–4 ¹	High = negative	2.5 ± 0.9	2.6 ± 0.9	2.1 ± 0.8	**<0.001**
Work engagement	2	0–6 ²	High = positive	5.1 ± 0.8	5.1 ± 0.8	5.0 ± 0.9	0.46
Psychosocial safety climate	10	1–5 ²	High = positive	27.8 ± 9.0	27.0 ± 8.8	31.0 ± 9.0	**<0.001**
Management commitment	11	1–5 ²	High = positive	7.5 ± 2.6	7.3 ± 2.6	8.4 ± 2.4	**<0.001**
Management priority	8	1–5 ²	High = positive	7.3 ± 2.7	7.1 ± 2.6	8.2 ± 2.7	**<0.001**
Organizational communication	12	1–5 ²	High = positive	6.1 ± 2.5	5.8 ± 2.4	7.1 ± 2.3	**<0.001**
Organizational participation	14	1–5 ²	High = positive	7.2 ± 2.5	7.1 ± 2.5	7.8 ± 2.5	**0.010**

Note. ^a^ One individual answered “Other” in sex and was not included in the stratified analyses. ¹Higher scores indicate a more unfavorable situation. ² Higher scores indicate a more favorable situation. Mean = the mean value of the total score for respective instrument; sd = the standard deviation of the mean value; *n* = the number of individuals. * The *p*-values show the difference between female and male teachers, calculated by the T-test and indicating how consistent the results are with the 0-hypothesis with a significance limit of 0.05; bold figures representing *p*-values < 0.05.

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
