# Peer review of "Health and Work Environment among Female and Male Swedish Elementary School Teachers—A Cross-Sectional Study"

_ijerph, 2019, doi:10.3390/ijerph17010227_

Round 1

Reviewer 1 Report

Dear Authors and Editor,

the manuscript "Health and Work Environment Among Female and Male Swedish Elementary School Teachers - a Cross4 Sectional Study" presents interesting data on work-related psychosocial risks and possible health consequences among elementary school teachers in Sweden: major revisions are needed before the article could be considered ready for publication.

First of all, I do not agree on the inclusion of the Burn ou among the "common mental disorders". According to the recent WHO definition (ICD-11) https://www.who.int/mental_health/evidence/burn-out/en/  “Burn-out is a syndrome conceptualized as resulting from chronic workplace stress that has not been successfully managed. It is characterized by three dimensions:

feelings of energy depletion or exhaustion; increased mental distance from one’s job, or feelings of negativism or cynicism related to one's job; and reduced professional efficacy.

Burn-out refers specifically to phenomena in the occupational context and should not be applied to describe experiences in other areas of life.”

Accordingly, it can not be defined as a mental disorder, even if burn-out syndrome may have health consequences for the workers associated to some mental disorders, and it has to be noted that burn-out is not included in the Diagnostic and Statistical Manual of Mental Disorders , the main recognized tool applied for the categorization of mental illnesses diagnosis.

I suggest to separate in the introduction burn-out from CMDs. E.g.  "In Sweden, they have been identified as one of professions with the highest prevalence of CMDs" should be " In Sweden, they have been identified as one of professions with the highest prevalence of CMDs and Burn-out".

In general, the attention given by the Authors to CMDs and Burn-out in the introduction is definitely too much, as the manuscript does not focus on these aspects, but only on perceived health status in general, distress, work sickness absence and safety climate at the workplace: I suggest to shorten the introduction and not focusing too much into mental health, which is not a result of the study.

In the material and method section some important information on response rates and reasons for not participating in the study of teachers who refused to participate (if any) should be described.

In the results section Table 1 has too much data for the description of the sample, not used in further analysis in the paper, and accordingly not useful for the reader (e.g. education level, ordinary working time ).

Discussion: Lines 26-42: I warmly suggest to fully remove this not scientific political discussion, whoich is out of the purposes of a scientific paper published in this kind of journals.

Conclusions: I suggest to delete lines 120-122 as this study does not allow to conclude anything on CMDs, and to reformulate lines 124-126.

Best regards,

the Reviewer.

Author Response

We would to thank the reviewer for the complements given. We have addressed the points raised accordingly below.

1. First of all, I do not agree on the inclusion of the Burn ou among the "common mental disorders. I suggest to separate in the introduction burn-out from CMDs. E.g.  "In Sweden, they have been identified as one of professions with the highest prevalence of CMDs" should be " In Sweden, they have been identified as one of professions with the highest prevalence of CMDs and Burn-out".

Thank you for this comment, we have throughout the introduction separated burn-out from CMDs, for example page 3 lines 53, 58, 67 and 71.

2. In general, the attention given by the Authors to CMDs and Burn-out in the introduction is definitely too much, as the manuscript does not focus on these aspects, but only on perceived health status in general, distress, work sickness absence and safety climate at the workplace: I suggest to shorten the introduction and not focusing too much into mental health, which is not a result of the study.

We have shortened the introduction as much as we though possible, to still remain the clarity in the introduction that we are examining risk and protective factors in teachers work environment that are associated with CMDs and burnout.

3. In the material and method section some important information on response rates and reasons for not participating in the study of teachers who refused to participate (if any) should be described.

Thank you for this comment. We have added additional information on response rates (page 4, lines 149 and 152-153).

4. In the results section Table 1 has too much data for the description of the sample, not used in further analysis in the paper, and accordingly not useful for the reader (e.g. education level, ordinary working time ).

As suggested by the reviewer we have deleted the following information from table 1: education level, ordinary working time and children under a certain age.

5. Discussion: Lines 26-42: I warmly suggest to fully remove this not scientific political discussion, whoich is out of the purposes of a scientific paper published in this kind of journals.

Thank you for this comment. We agree with the reviewer that we in this part of the manuscript refer to political reforms. We have however chosen to keep this discussion, as we believe that that the political reforms that have occurred over many years in Sweden give possible explanations for the findings of the study.

6. Conclusions: I suggest to delete lines 120-122 as this study does not allow to conclude anything on CMDs, and to reformulate lines 124-126.

In accordance with the reviewer’s suggestion we have deleted lines 120-122. We have also accordingly reformulated lines 124-126.

Reviewer 2 Report

Thank you for the opportunity to review this manuscript. This article provides a good descriptive analysis of factors that may impact upon teachers mental health. The data collection methods are described well, as are the results. The tables are clear and easy to read and the discussion is appropriately brought back to the literature. This study adds new knowledge to our understanding of factors that may impact teachers well-being in the work place.

Author Response

We would to thank the reviewer for the complements given.

Reviewer 3 Report

The manuscript is original and relevant. The findings are interesting and bring insights about possible intervention strategies. I have minor points to highlight: 1. It should be relevant for readers to know which questionnaires were used in the abstract or keywords; 2. The information about the overall distribution of female teachers in elementary schools in Sweden (in the Discussion section) could appear in the Introduction section (when authors discuss different rates of sick leave between gender); 3. Authors should discuss if the unequal distribution between male and female teachers affect the findings in any sense; 4. The questionnaire asks about sex and at the Discussion section gender differences are presented and discussed. One subject answered "other sex" and was excluded. I am intrigued to know the authors' opinion about the use of "sex" question instead of "gender" question in studies like that. I would be happy to read something about it in this paper.

Author Response

We would to thank the reviewer for the complements given. We have addressed the points raised accordingly below.

It should be relevant for readers to know which questionnaires were used in the abstract or keywords;

Thank you for this comment. We have now accordingly added the questionnaires in the abstract (page 1, lines 36-37)

The information about the overall distribution of female teachers in elementary schools in Sweden (in the Discussion section) could appear in the Introduction section (when authors discuss different rates of sick leave between gender);

We have now accordingly moved the sentence regarding the overall distribution of female teachers to the Introduction section (page 2, lines 62-63).

Authors should discuss if the unequal distribution between male and female teachers affect the findings in any sense;

We agree with the reviewer that this should be discussed. We have added a sentence regarding this in the discussion (lines 474-476) describing that we do not think that the unequal distribution has affected the findings.

The questionnaire asks about sex and at the Discussion section gender differences are presented and discussed. One subject answered "other sex" and was excluded. I am intrigued to know the authors' opinion about the use of "sex" question instead of "gender" question in studies like that. I would be happy to read something about it in this paper.

Thank you for this interesting and important issue raised. In the Swedish language there is only one word for sex and gender, i.e. the same word is used for both gender and sex. In the survey used in this study we asked participants whether they were woman, man or other. The third option “other” was included to allow participants to provide a response even if they do not identify as either woman or man, or if they prefer not to respond. We have added this clarification to the method section of the manuscript.

Round 2

Reviewer 1 Report

Congratulations to the Authors: the manuscript has been significantly improved and it is now ready for publication.